# Urinary microRNAs as Prognostic Biomarkers for Predicting the Efficacy of Immune Checkpoint Inhibitors in Patients with Urothelial Carcinoma

**DOI:** 10.3390/cancers17162640

**Published:** 2025-08-13

**Authors:** Yosuke Hirasawa, Atsushi Satomura, Mitsuo Okada, Mieko Utsugi, Hiroki Ogura, Tsuyoshi Yanagi, Yuta Nakamori, Masayuki Takehara, Kokichi Murakami, Go Nagao, Takeshi Kashima, Naoya Satake, Yoriko Ando, Motoki Mikami, Mika Mizunuma, Yuki Ichikawa, Yoshio Ohno

**Affiliations:** 1Department of Urology, Tokyo Medical University, 6-7-1 Nishi-shinjuku, Shinjuku-ku, Tokyo 160-0023, Japan; wbqmd473@yahoo.co.jp (Y.H.); mitsuwo.okada.uro@gmail.com (M.O.); hirokiogura.214@gmail.com (H.O.); yanagituyoshi@gmail.com (T.Y.); yuta90790279@gmail.com (Y.N.); take3.3keta@gmail.com (M.T.); komurakami0914@gmail.com (K.M.); go.nagao@gmail.com (G.N.); takeshikashima922@yahoo.co.jp (T.K.); satake@tokyo-med.ac.jp (N.S.); 2Craif. Inc., Nagoya 464-0814, Aichi, Japan; atsushi.satomura@craif.com (A.S.); yoriko.ando@craif.com (Y.A.); motoki.mikami@craif.com (M.M.); mika.mizunuma@craif.com (M.M.); yuki.ichikawa@craif.com (Y.I.); 3Institute of Innovation for Future Society, Nagoya University, Nagoya 464-8601, Aichi, Japan; 4The Center for Diversity at Tokyo Medical University, 6-7-1 Nishi-shinjuku, Shinjuku-ku, Tokyo 160-0023, Japan; tm00220@tokyo-med.ac.jp

**Keywords:** miRNA, urothelial carcinoma, bladder cancer, immune check point inhibitors, biomarker

## Abstract

Immune checkpoint inhibitors (ICIs) have transformed urothelial carcinoma (UC) treatment, yet only a minority of patients benefit long-term owing to variable response rates (<25%) and emerging resistance. This study investigates urinary microRNAs (miRNAs) as noninvasive predictive biomarkers for ICI efficacy. Urine samples collected before initiation of ICI were analyzed via next-generation sequencing. Patients were grouped into responders (those with disease control for ≥6 months) or non-responders (those with disease progression within 6 months). Urinary levels of miR 185 5p and miR 425 5p were significantly elevated in responders, correlating with enhanced overall and progression-free survival (*p* < 0.05). Conversely, miR 30a 5p and miR 542 3p were higher in non-responders, suggesting a potential role in immune resistance. Therefore, these urinary miR 185 5p and miR 425 5p can serve as favorable predictive biomarkers for ICI response, while miR 30a 5p and miR 542 3p could be a signal of resistance for ICI treatment. Furthermore, urine-based miRNA profiling is a promising tool to inform personalized immunotherapy strategies in UC management.

## 1. Introduction

Urothelial carcinoma (UC) is the most common malignancy of the urinary tract and has a high recurrence rate and significant mortality in advanced stages [1]. Although platinum-based chemotherapy has been the mainstay of treatment for metastatic UC, the emergence of immune checkpoint inhibitors (ICIs) has revolutionized therapeutic strategies, offering durable responses in a subset of patients [2,3]. However, the overall efficacy of ICIs in UC remains limited, with response rates generally <25% in unselected patient populations [2,3]. Primary and acquired resistance to ICIs remains a significant challenge, necessitating the identification of predictive biomarkers and novel therapeutic targets.

The objective response rate (ORR) of ICIs for UC remains low, and a significant proportion of patients fail to derive substantial clinical benefits. Identifying patients who will respond to ICI therapy and those who will experience early resistance is critical for optimizing treatment strategies. Biomarkers capable of predicting ICI responses are urgently required to guide patient selection and improve therapeutic outcomes. Several studies have explored potential biomarkers, including programmed death-ligand 1 (PD-L1) expression, tumor mutational burden (TMB), and gene expression signatures; however, none have demonstrated consistent predictive value across clinical trials [2,3,4,5,6,7]. The development of reliable biomarkers is essential for stratifying patients and personalizing immunotherapeutic approaches to UC.

Pembrolizumab (Pem), an anti-programmed death 1 (PD-1) monoclonal antibody, has been approved for the treatment of advanced or metastatic UC following the failure of platinum-based chemotherapy [2]. Despite its promising mechanism of action, clinical trials have demonstrated its limited efficacy in UC. The KEYNOTE-045 trial reported an ORR of only 21.1% in the Pem-treated cohort, with a median overall survival (OS) of 10.3 months compared with 7.4 months with chemotherapy [2]. Furthermore, a significant proportion of patients do not derive long-term benefits, and immune-related adverse events present additional treatment challenges [8].

Avelumab (Ave), an anti-PD-L1 monoclonal antibody, has been approved as a maintenance therapy for advanced UC following platinum-based chemotherapy [6,9]. The JAVELIN Bladder 100 trial demonstrated an improved median OS of 21.4 months with Ave maintenance compared with 14.3 months with best supportive care alone [9]. However, the overall response rate is not sufficient, and a significant proportion of patients experience disease progression despite treatment. Additionally, real-world data suggest that many patients cannot receive Ave maintenance because of rapid disease progression following chemotherapy, limiting its clinical utility [10].

Nivolumab (Nivo), an anti-PD-1 monoclonal antibody, has been investigated as adjuvant therapy for patients with high-risk muscle-invasive UC after radical surgery [7]. The CheckMate 274 trial demonstrated that adjuvant Nivo prolonged disease-free survival compared with the placebo group; however, its impact on OS remains unclear [7]. Furthermore, a significant proportion of patients experience disease recurrence despite treatment, raising concerns about long-term benefits [7]. Additionally, biomarkers, such as PD-L1 expression, do not consistently predict treatment responses, highlighting the need for better patient stratification [11].

MicroRNAs (miRNAs) are small non-coding RNAs that regulate gene expression at the post-transcriptional level, playing crucial roles in cancer pathogenesis, including tumor proliferation, invasion, and immune evasion [12,13]. Preclinical models have demonstrated tumor suppression via miR-34a restoration or oncomiR inhibition [14,15,16]. Nonetheless, despite robust preclinical efficacy, clinical translation of miRNA therapies has encountered barriers such as poor pharmacokinetics, delivery inefficiency, off-target effects, immunogenicity, and scale-up issues [14,15,16,17,18,19]. In UC, dysregulated miRNA expression is associated with disease progression, treatment resistance, and immune modulation [20,21]. Specific miRNAs regulate key components of the immune checkpoint pathway, including PD-L1 expression and T-cell exhaustion [22,23]. Given their roles in immune regulation, miRNAs have garnered attention as potential predictive biomarkers for ICI efficacy.

Recent studies have suggested that specific miRNAs can modulate the tumor microenvironment by influencing immune cell infiltration, cytokine production, and antigen presentation, thereby affecting ICI treatment responses [24,25]. While most existing studies have focused on blood-based miRNAs, urinary miRNAs also reflect tumor- and immune-related changes in UC [26]. The potential to capture immune activation or suppression in the tumor microenvironment makes urinary miRNAs promising candidates for predicting ICI response.

Despite these insights, the clinical application of miRNA-based biomarkers remains in its early stages, and further studies are required to validate their predictive value in patients with UC receiving ICI therapy.

This study aimed to explore the potential of urinary miRNAs as predictive and non-invasive biomarkers for assessing the therapeutic efficacy of ICIs in UC patients.

## 2. Materials and Methods

### 2.1. Patients and Samples

The institutional review board (T2021-0278) approved the current prospective study, and this study complied with the 1964 Declaration of Helsinki and its later amendments. We identified 12 patients with advanced UC who received one of the ICIs (Pem, Ave, and Nivo) at our institution between December 2022 and June 2023 (Figure 1). Histopathological examination confirmed UC in all enrolled patients. Patients exhibiting radiologically confirmed progressive disease (PD) following first-line chemotherapy were administered Pem (Figure 1). In contrast, those with stable disease (SD), partial response (PR), or complete response (CR) based on radiological assessments received Ave as maintenance treatment. Patients at high risk of recurrence—defined as having a pathological stage of pT3, pT4a, or pN+, or ypT2 to ypT4a or ypN+ following neoadjuvant cisplatin-based chemotherapy—were treated with Nivo after undergoing total cystectomy (Figure 1). Radiological evaluations were routinely conducted using computed tomography (CT) before and after every three cycles of ICI therapy. Additional assessments were performed at the discretion of the treating physicians, particularly when clinical symptoms deteriorated. Tumor response was assessed according to the Response Evaluation Criteria in Solid Tumors version 1.1 [27]. The best response for each metastatic lesion was categorized as follows: (1) CR, defined as the disappearance of lesions or a reduction in short-axis diameter of all lymph node metastases to less than 10 mm; (2) PR, indicated by a reduction of more than 30%; (3) SD, not meeting criteria for CR, PR, or PD; and (4) PD, defined as an increase in lesion size exceeding 20%. Pem (200 mg) was administered intravenously every 3 weeks, and Ave (800 mg) was administered intravenously every 2 weeks following first-line chemotherapy, with treatment continuing until disease progression. Patients received Nivo at 240 mg via 30 min intravenous infusion every 2 weeks, with treatment continued until disease recurrence or progression on imaging. Urine samples were collected from patients after they received an explanation of the study and provided informed consent, prior to the initiation of ICI therapy. The samples were promptly refrigerated at 4 °C, aliquoted within 72 h, and then stored at −80 °C until further analysis.

We defined the “nonresponder group” as patients who had disease progression within 6 months after the initiation of ICI treatments and the “responder group” as those who had SD, PR, or CR > 6 months after the initiation of the ICI treatments. We explored the differences and the profile of urine miRNAs between the “nonresponder” and “responder” groups.

### 2.2. Isolation of Extracellular Vesicles from Urine and RNA Extraction

Approximately 3.5 mL of urine was centrifuged at 2000× *g* for 30 min at 4 °C to remove cells and debris. Next, 3 mL of the resulting supernatant was combined 1:1 with the Total Exosome Isolation Reagent (from urine) (Thermo Fisher Scientific, Waltham, MA, USA) and incubated at room temperature for 1 h with gentle shaking (100 rpm). The mixture was then centrifuged at 3000× *g* for 1 h at 4 °C to pellet extracellular vesicles. After the supernatant was discarded, the pellet was resuspended in 250 μL of phosphate-buffered saline. RNA was extracted from this suspension using the MagMAX mirVana Total RNA Isolation Kit (Thermo Fisher Scientific) on a KingFisher Apex System (Thermo Fisher Scientific), following the manufacturer’s guidelines. The initial elution volume was 30 μL of elution buffer. To concentrate the RNA, the solution was placed in a centrifugal concentrator (Eppendorf, Hamburg, Germany) at 60 °C for 30–45 min and re-eluted in 5 μL of nuclease-free water (Thermo Fisher Scientific). The resulting RNA extracts were stored at −80 °C until further use.

### 2.3. Small RNA Library Preparation and Sequencing

Small RNA libraries were constructed from 5 μL of RNA using the QIAseq miRNA Library Kit (QIAGEN, Hilden, Germany), following the manufacturer’s instructions. Library concentrations were assessed using the Qubit dsDNA HS Assay Kit (Thermo Fisher Scientific) on a Qubit Flex Fluorometer (Thermo Fisher Scientific), after which the libraries were stored at −20 °C. Sequencing was performed using an Illumina NextSeq 550 System (San Diego, CA, USA), generating 75-nucleotide single-end reads, in accordance with the manufacturer’s recommendations.

### 2.4. Processing of Small RNA Sequencing Reads

Raw sequencing reads were processed using unique molecular identifier (UMI) tools [28]. This step extracted the miRNA-specific 5′-end bases preceding the 3′-adaptor sequence and incorporated a 12 bp UMI located after the 3′-adaptor into each read identifier. Reads longer than 19 nucleotides were then aligned to the human miRNA reference dataset included in miRge 3.0 [29] using Bowtie v1.2.3 [30], permitting zero mismatches within a 25-nucleotide seed region and disallowing reverse complement alignment. miRNAs detected at a minimum of two counts in at least 80% of the samples were normalized to counts per million and included in subsequent analyses.

### 2.5. Differential Expression Analysis

Differentially expressed miRNAs were identified using DESeq2 [31], applying an adjusted *p*-value cutoff of 0.1. For inclusion in the analysis, a given miRNA was required to have at least two counts in ≥80% of the samples. Pathway enrichment was performed using the miRWalk [32]. Differential expression analysis accounting for the influence of hematuria was performed with DESeq2, using the presence or absence of hematuria as a covariate. To further correct hematuria-related profile variation, limma removeBatchEffect was used [33].

### 2.6. Urine Analysis

AUTION Sticks 10EA (Arkay, Kyoto, Japan) was used for urine analysis. Occult blood levels of − and ± were classified as without occult blood, whereas levels of 1+, 2+, and 3+ were classified as with occult blood.

### 2.7. Statistical Analyses

The *p* values were adjusted for multiple comparisons using the Benjamini–Hochberg method. Kaplan–Meier analysis and the log-rank test were used to compare progression-free survival (PFS) between the two groups, respectively. All statistical analyses were performed using R 4.3.1 for Windows (R Foundation for Statistical Computing, Vienna, Austria; http://www.r-project.org/). All parameters in this study with *p* < 0.05 were considered statistically significant.

## 3. Results

### 3.1. Patient Demographics

In this study, 12 patients with UC were enrolled, of whom 5 were nonresponders (NR) and 7 were responders (R) to ICIs (Table 1). Urinary occult blood of ≥1+ was observed in three patients from the R group and four patients from the NR group. More than 10^4^ total miRNAs were obtained from the R and NR groups (Appendix A). A clear difference in PFS was observed between the R and NR groups, with median PFS of 15.1 months in the R group and 3.7 months in the NR group (Appendix A).

### 3.2. miRNA Profiles

Principal component analysis of the miRNA profiles revealed a clear separation between the R and NR groups (Figure 2a). Differential expression analysis identified six miRNAs upregulated in the R group and four miRNAs upregulated in the NR group (Figure 2b,c). Pathway analysis of the target mRNAs of miRNAs upregulated in the R group identified four enriched pathways, whereas the analysis of miRNAs upregulated in the NR group identified seven enriched pathways (Appendix A). Because blood contamination can introduce blood cell-derived miRNAs that may confound urinary miRNA profiles, differential expression analysis was performed to compare samples with and without detectable occult blood to assess the potential influence of blood-derived miRNAs (Appendix A). Five miRNAs were upregulated in samples with occult blood, whereas one miRNA was downregulated. Among the 10 differentially expressed miRNAs in R, miR-486-5p and miR-23a-3p were upregulated in samples with occult blood, suggesting that occult blood was a confounding factor in this cohort and needed to be carefully controlled. Therefore, a further differential expression analysis with occult blood as a covariate was conducted. miR-486-5p, miR-210-3p, and miR-151a-3p, which had shown increased expression in R, and miR-542-3p and miR-20a-5p, which had shown decreased expression (Figure 2b), remained differentially expressed in the same direction in R (Appendix A). Furthermore, after adjusting for the effect of occult blood, a clear shift between R and NR was still observed in the miRNA profile along PC3 (Appendix A). Collectively, these results suggest that urinary miRNA profiles are promising predictive biomarkers for ICI response.

### 3.3. PFS Analysis

We investigated the potential of the six miRNAs that were upregulated in the R group (Figure 2b) as biomarkers. High expression levels of miR-186-5p and miR-425-5p were significantly associated with longer PFS (Figure 3). In the high-expression group of miR-186-5p, the median PFS was 15.0 months, compared with 3.27 months in the low-expression group. Similarly, the high-expression group of miR-425-5p had a median PFS of 15.0 months, whereas it was 3.27 months in the low-expression group. We also examined the relationship between PFS and the four miRNAs upregulated in the NR group. Among these, miR-30a-5p and miR-542-3p showed significant differences in PFS (Figure 4). In the high-expression group of miR-30a-5p, the median PFS period was 3.27 months, compared with 15.0 months in the low-expression group (Figure 4). Similarly, the high-expression group of miR-542-3p had a median PFS of 3.95 months, whereas the low-expression group had a median PFS of 15.0 months (Figure 4).

## 4. Discussion

To the best of our knowledge, this is the first study investigating the correlation between the efficacy of ICIs and urinary miRNAs in patients with advanced metastatic or muscle-invasive UC. In the current study, we found that miR-186-5p and miR-425-5p were significantly upregulated in urine samples from the R group, whereas miR-30a-5p and miR-542-3p were significantly upregulated in urine samples from the NR group, which significantly correlated with PFS.

The limited efficacy of ICIs for UC underscores the urgent need for predictive biomarkers to optimize patient selection and treatment strategies [2,3,6,7,8,9,10,11]. The low ORR of Pem and Ave [2,6,9,10] and the finding that 25% of patients experienced recurrence or metastasis within just 6 months of adjuvant Nivo therapy [7,11] in clinical trials highlight that only a subset of patients benefit from immunotherapy, whereas many experience disease progression despite treatment.

Current biomarkers, such as PD-L1 expression and TMB, have shown inconsistent predictive values, reinforcing the need for novel approaches for patient stratification [2,3,4,5,6,7]. In the latest version of the miRBase database (v22), approximately 2600 mature human miRNAs are registered [34]. These miRNAs exhibit distinct expression patterns depending on the cancer type and treatment modality [34]. miRNAs have emerged as promising candidates for predicting ICI response owing to their regulatory roles in immune modulation and tumor microenvironment interactions [35,36,37,38]. Several studies have identified miRNAs associated with PD-L1 expression, T-cell exhaustion, and immune evasion mechanisms, suggesting their potential as biomarkers for treatment efficacy [39,40]. However, challenges remain in translating these findings into clinical practice, including the need for standardized methodologies and validation in prospective trials.

Preclinical studies have suggested that modulating specific miRNAs can improve immune checkpoint blockade responses; however, further studies are required to establish their clinical relevance [41]. Cheng et al. reported that miR-186-5p acted as a tumor suppressor and inhibited PD-L1 expression in cancer cells and miR-186-5p directly targeted molecules involved in the immune response [42]. Restoring miR-186-5p expression has been suggested as a potential therapeutic strategy for enhancing the response to PD-1/PD-L1 inhibitors. By reducing PD-L1 expression and promoting T-cell activation, miR-186-5p can improve the efficacy of PD-1 blockade therapies and counteract the immune evasion mechanisms employed by tumors. miR-186-5p plays a dual role, acting as a tumor suppressor and a modulator of immune cell function. In the current study, it was demonstrated that miR-186-5p was significantly upregulated in urine samples from the R group. Based on these previous reports and the results of our study, miR-186-5p could have potential as a biomarker for predicting the favorable efficacy of ICI treatment in patients with UC.

miR-186-5p is downregulated in bladder cancer tissues, and its expression is inversely correlated with tumor grade and stage [43]. miR-186-5p inhibits UC cell proliferation and induces apoptosis by targeting key oncogenes, such as BCL2 and MAPK1. Additionally, miR-186-5p is involved in the regulation of cancer cell migration and invasion, and miR-186-5p promotes metastasis by regulating epithelial-to-mesenchymal transition (EMT) markers, such as E-cadherin and vimentin. The downregulation of miR-186-5p leads to the upregulation of EMT-related transcription factors, facilitating the invasive potential of cancer cells [43]. Furthermore, its modulatory effects on key cancer-related pathways suggest that miR-186-5p is a promising therapeutic target. Particularly, miR-186-5p is a potential tumor suppressor in certain cancers. For example, its downregulation is associated with increased cell proliferation and reduced apoptosis in breast cancer cell lines, implying that miR-186-5p would serve as a protective factor against tumorigenesis [44]. Furthermore, miR-186-5p targets key oncogenes involved in cell cycle regulation, such as cyclin D1, suggesting a role in modulating cell cycle progression [45]. These findings suggested its potential as both a diagnostic and therapeutic target in UC, in addition to being a predictive biomarker for the efficacy of ICI treatment. However, despite extensive preclinical research, clinical translation of miRNA-based therapies has faced significant challenges and limited success. The liposomal miR-34a mimic MRX34 was the first miRNA therapeutic in a phase 1 clinical trial (NCT01829971), but development was terminated due to severe immune-mediated toxicities and patient deaths [16,46]. Similarly, trials of other miRNA mimics, including miR-29b for fibrosis and several antimiRs targeting miR-21 and miR-122, were discontinued due to toxicity or lack of efficacy [17,47]. It has been reported that difficult challenges in advancing miRNA therapeutics include poor pharmacokinetics, off-target effects, and inadequate delivery specificity [16,47,48].

It was reported that miR-30a-5p was involved in resistance to anti-PD-1 immunotherapy in non-small cell lung cancer [48]. This resistance is induced by has_circ_0020714, which acts as a sponge for miR-30a-5p, thereby promoting the expression of SOX4 [48]. These findings suggest that miR-30a-5p influences the efficacy of cancer immunotherapy. Additionally, patients with lung adenocarcinoma with low expression of miR-30a-5p exhibit increased infiltration of B cells, CD4+, T cells, and macrophages, and this infiltration of immune cells is associated with improved patient prognosis [49]. Based on the role of miR-30a-5p in the regulation of immune responses, its ability to modulate inflammatory pathways and immune cell functions highlights its potential as a biomarker for predicting the effects of ICIs. In our study, miR-30a-5p was significantly upregulated in urine samples from the NR group, which significantly correlated with worse PFS. In addition to the previous reports regarding the correlation between the efficacy of ICI treatment and lung cancer, miR-30a-5p could be a potential predictive biomarker for the efficacy of ICI in UC patients.

miR-425-5p contributes to immune evasion in esophageal cancer cells by suppressing necroptosis, a form of programmed cell death. Specifically, it promotes the expression of BCAT1 and is associated with an increased expression of PD-L1 (CD274) [50]. As PD-L1 expression is upregulated by miR-425-5p, tumors with high miR-425-5p levels could be more responsive to anti-PD-L1 therapies.

However, there is currently no clear evidence that high miR-542-3p expression is directly associated with cancer therapy resistance. To the best of our knowledge, this is the first study to report an association between high miR-542-3p expression and immunotherapy resistance in patients with UC.

Based on the result of the current study, miR-185-5p and miR-425-5p can serve as predictive biomarkers of favorable ICI efficacy in patients with UC, whereas miR-30a-5p and miR-542-3p could be associated with resistance mechanisms for ICI treatment.

There were some limitations in the current study. This study includes just 12 patients. Further research with a larger cohort is needed to confirm the generalizability of these biomarkers. Validation in a larger cohort will also be required to confirm the predictive value of the identified urinary miRNAs and the clinical utility of our findings. However, we believe that the accumulation of such exploratory studies represents an essential first step. Based on the results of the present study, we aim to progressively increase the number of cases and ultimately conduct more extensive investigations in the future. In addition, occult blood acts as a potential confounder by introducing blood-derived miRNAs into urine samples in our cohort. Although we have adjusted for the confounding effect, residual bias cannot be completely ruled out.

## 5. Conclusions

Although ICIs have provided a breakthrough in UC treatment, their limited efficacy requires the identification of robust predictive biomarkers. We found that miR-186-5p and miR-425-5p were significantly upregulated in the urine of the R group, whereas miR-30a-5p and miR-542-3p were significantly upregulated in the urine of the NR group. These miRNAs were also significantly correlated with PFS in patients with UC who received ICI treatment. These results showed that urinary miR 185 5p and miR 425 5p can serve as favorable predictive biomarkers for ICI response, while miR 30a 5p and miR 542 3p could be a signal of resistance for ICI treatment. Urinary miRNAs can be obtained using a minimally invasive procedure and would serve as useful biomarkers for predicting the efficacy of immunotherapy in patients with UC. miRNAs could have potential for refining patient selection and guiding therapeutic decisions; however, additional large-scale studies are needed to confirm their clinical utility in the future.

## Figures and Tables

**Figure 1 cancers-17-02640-f001:**
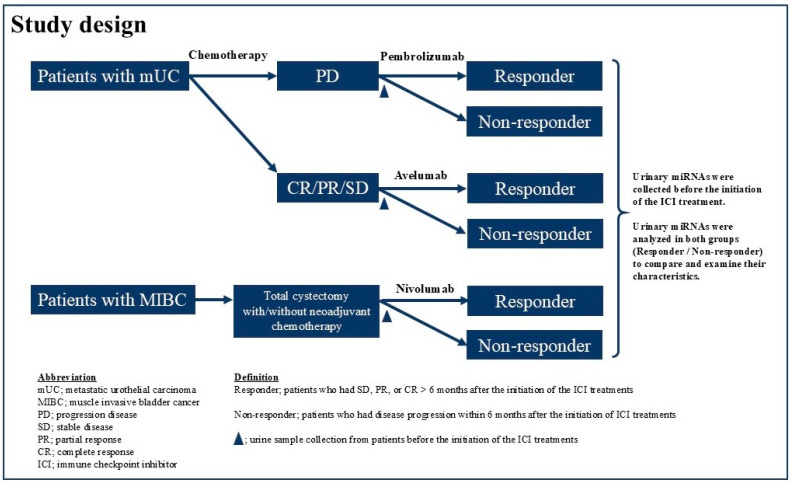
Study design.

**Figure 2 cancers-17-02640-f002:**
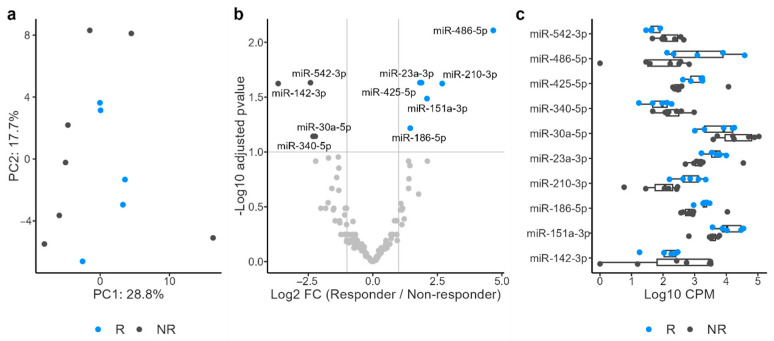
Principal component analysis of microRNA (miRNA) profiles revealed a clear separation between responders (R) and nonresponders (NR) (**a**). Differential expression analysis identified six miRNAs upregulated in R and four miRNAs upregulated in NR (**b**,**c**).

**Figure 3 cancers-17-02640-f003:**
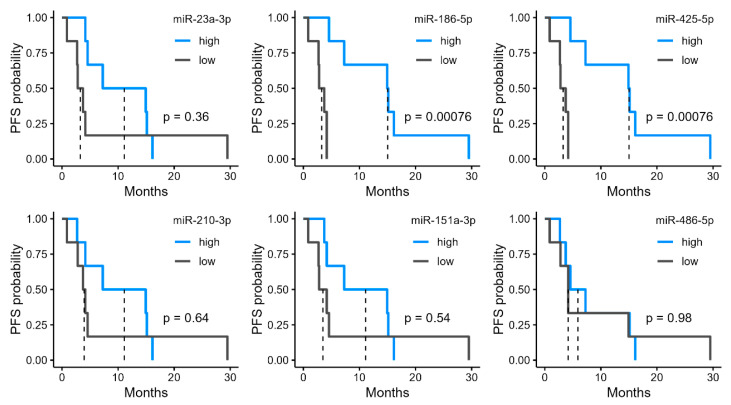
Progression-free survival (PFS) stratified by urinary microRNA (miRNA) levels upregulated in responders (R). Kaplan–Meier curves of PFS in patients grouped into high and low expression by median levels of six urinary miRNAs identified as significantly upregulated in R.

**Figure 4 cancers-17-02640-f004:**
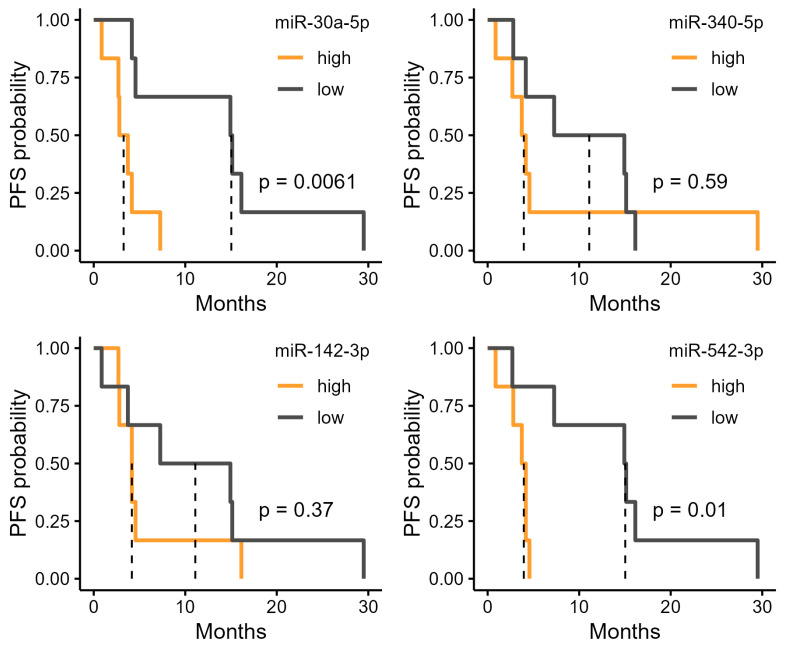
Progression-free survival (PFS) stratified by urinary microRNA (miRNA) levels upregulated in non-responders (NR). Kaplan–Meier curves of PFS in patients grouped into high and low expression by median levels of six urinary miRNAs identified as significantly downregulated in responders (R).

**Table 1 cancers-17-02640-t001:** Patient characteristics.

	Responder N = 5 ^1^	Non-Responder N = 7 ^1^
**Age, mean ± SE (years)**	78.8 ± 3.8	74.3 ± 2.8
**Sex**		
Male	4 (80%)	6 (85%)
**Height, mean ± SE (cm)**	162.6 ± 5.0	163.7 ± 4.1
**Weight, mean ± SE (kg)**	56.3 ± 7.6	58.3 ± 3.9
**Body mass index, mean ± SE (kg/m^2^)**	21.2 ± 2.2	21.8 ± 1.2
**Cancer**		
Urothelial	5 (100%)	7 (100%)
**ICI**		
Avelumab	1 (20%)	1 (14%)
Nivolumab	1 (20%)	3 (43%)
Pembrolizumab	3 (60%)	3 (43%)
**Stage**		
III	0 (0%)	1 (14%)
IIIa	2 (40%)	0 (0%)
IIIb	0 (0%)	2 (29%)
IV	3 (60%)	4 (57%)
**Progressive disease**		
Non-PD	4 (80%)	0 (0%)
PD	1 (20%)	7 (100%)
**ICI month**	7.5 (5.0–7.8)	2.8 (2.3–4.2)
**PFS month**	15.1 (14.9–16.1)	3.7 (2.7–4.2)
**Occult blood**		
-	1 (20%)	3 (43%)
±	1 (20%)	0 (0%)
1+	0 (0%)	3 (43%)
2+	1 (20%)	0 (0%)
3+	2 (40%)	1 (14%)

^1^ Mean ± SE; n (%); median (Q1–Q3), SE; standard error.

## Data Availability

The data from this study can be accessed by academic and commercial partners upon reasonable request, subject to Institutional Review Board approval and a data use agreement. For further details or reanalysis of the study data, contact Yosuke Hirasawa at wbqmd473@yahoo.co.jp.

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
