# Peer review of "Urinary microRNAs as Prognostic Biomarkers for Predicting the Efficacy of Immune Checkpoint Inhibitors in Patients with Urothelial Carcinoma"

_cancers, 2025, doi:10.3390/cancers17162640_

Round 1

Reviewer 1 Report

Comments and Suggestions for Authors

The authors concentrated on the urinary microRNAs (miRNAs) as a means of detecting the response to immune checkpoint inhibitors (ICI) in urothelial carcinomas (UC). In results, high expression of miR-185-5p and miR-425-5p and low expression of miR-30a-5p and miR-542-3p were detected in ICI response UC. The expression of these microRNAs has been linked to progression-free survival (PFS). They found new and impressive biomarkers at UC using a new technique. However, there are some questions and suggestions as described below.

Major comments

  • Therefore, Given the congruence between the cases for biomarker detection and for verifying PFS based on biomarker expression, confirming the certainty of its benefit has proven to be a formidable challenge. The authors should consider augmenting the cases with additional examples to enhance the comprehensiveness of the PFS analysis.

Minor comments

  • The authors should write the formal name of “SD” on page 3, line 129.
  • In Table 1, the authors should present the standard error of the mean (SEM) for the variables of age, height, weight, and BMI.
  • The authors should write the formal name of “BMI” in Table 1.

Author Response

Comments to the Author
Major comments

Therefore, Given the congruence between the cases for biomarker detection and for verifying PFS based on biomarker expression, confirming the certainty of its benefit has proven to be a formidable challenge. The authors should consider augmenting the cases with additional examples to enhance the comprehensiveness of the PFS analysis.

Response:

We are thankful to the Reviewer’s comment. As pointed out by Reviewer 1, this study includes the small number of patients (12 patients), causing the weakness of statistical power and clinical utility. However, unfortunately, it was difficult to add a greater number of patients during this short period. Thus, we added the limitation and sentence “This study includes just 12 patients. Further research with a larger cohort is needed to confirm the generalizability of these biomarkers. Validation in a larger cohort will be also required to confirm the predictive value of the identified urinary miRNAs and the clinical utility of our findings. However, we believe that the accumulation of such exploratory studies represents an essential first step. Based on the results of the present study, we aim to progressively increase the number of cases and ultimately conduct more extensive investigations in the future.” in the limitation area (page 16 line 1-7).

Minor comments

  • The authors should write the formal name of “SD” on page 3, line 129.

Response: We are thankful to the Reviewer’s comment. Based on the reviewer’s comment, I added the name of “SD” and added the sentence “stable disease (SD)” in page 7 line 10.

  • In Table 1, the authors should present the standard error of the mean (SEM) for the variables of age, height, weight, and BMI.

Response: We are thankful to the Reviewer’s comment. Based on the reviewer’s comment, we added the standard error of the mean for variables of age, height, weight, and BMI in Table 1 (page 19, 20)

  • The authors should write the formal name of “BMI” in Table 1.

Response: We are thankful to the Reviewer’s comment. Based on the reviewer’s comment, we added the formal name of BMI (Body mass index) in Table 1 (page 19)

Reviewer 2 Report

Comments and Suggestions for Authors

The authors have demonstrated the role of Urinary microRNAs as prognostic biomarkers. The presented work is important for urothelial carcinoma patients. However, before acceptance, the authors also improved the introduction and discussion sections. They should also mention how miRNAs can be used as therapy. So far, lots of research has been going on without much success. So the authors should mention the limitations of the study. The authors should also mention how sensitive the identified miRNAs are. 

Author Response

Comments to the Author
The authors have demonstrated the role of Urinary microRNAs as prognostic biomarkers. The presented work is important for urothelial carcinoma patients. However, before acceptance, the authors also improved the introduction and discussion sections. They should also mention how miRNAs can be used as therapy. So far, lots of research has been going on without much success. So the authors should mention the limitations of the study. The authors should also mention how sensitive the identified miRNAs are. 

Response; We are thankful to the Reviewer’s comment. As pointed out by Reviewer 2, we need to improve the introduction and discussion sections. And we need to mention the results of miRNA therapy and mention the limitations of the study. And we also mention how sensitive the identified miRNA are.

For introduction session, We added the sentence “Preclinical models have demonstrated tumor suppression via miR-34a restoration or oncomiR inhibition [16-18]. Nonetheless, despite robust preclinical efficacy, clinical translation of miRNA therapies has encountered barriers such as poor pharmacokinetics, delivery inefficiency, off-target effects, immunogenicity, and scale-up issues [16-22].” (page 5 line 17-21) and added some reference below. (page 30 line 27-35) (page 31 line 1-12)

  1. Xue, C.; Kim, T.; Croce, C.M.; Rossi, M.; Lee, J.; Zhang, Y.; Wang, L.; Chen, S.; Patel, D.; Singh, A.; et al. MicroRNA: trends in clinical trials of cancer diagnosis and therapy. Exp. Mol. Med. 2023, 55, 1012–1025. https://doi.org/10.1038/s12276-023-01050-9.
  2. D’Angelo, D.D.; Smith, H.A.; Karp, J.E.; Ross, C.; Fernandez, M.; Patel, R.; Johnson, B.; Lee, S.; Walker, C.; Green, M.; et al. Emergent miRNA therapeutics: from preclinical to clinic. Hum. Gene Ther. 2024, 35, 12–24.
  3. Beg, M.S.; Brenner, A.J.; Sachdev, J.; Borad, M.; Kang, Y.-K.; Stoudemire, J.; Smith, S.; Bader, A.G.; Kim, S.; Hong, D.S.; et al. Phase I study of MRX34, a liposomal miR‑34a mimic, administered twice weekly in patients with advanced solid tumors. Invest. New Drugs 2017, 35, 180–188. https://doi.org/10.1007/s10637-016-0407-y.
  4. He, X.H.; Liang, L.; Zhang, Y.; Wang, J.; Li, H.; Liu, W.; Chen, Y.; Zhang, X.; Li, J.; Wang, Z.; et al. A narrative review of microRNA therapeutics. Precis. Clin. Med. 2021, 4, 290–302. https://doi.org/10.21037/pcm-21-28.
  5. Seyhan, A.A. Trials and Tribulations of MicroRNA Therapeutics. Int. J. Mol. Sci. 2024, 25, 1469. https://doi.org/10.3390/ijms25031469.
  6. Di Fiore, R.; Tagliaferri, P.; Tassone, P.; De Luca, L.; Caraglia, M.; De Rosa, G.; Ciliberto, G.; Botti, G.; De Palma, R.; De Laurentiis, M.; et al. MicroRNA in cancer therapy: breakthroughs and challenges in early-stage development. J. Exp. Clin. Cancer Res. 2025, 44, 101. https://doi.org/10.1186/s13046-025-03391-X.
  7. What will it take to get miRNA therapies to market? Nat. Biotechnol. 2024, 42, 345–350. https://doi.org/10.1038/s41587-024-02480-0.

For discussion section, we added the sentence “In the current study, it was demonstrated that miR-186-5p was significantly upregulated in urine samples from the R group. Based on these previous reports and the result of our study, miR-186-5p could have potential as biomarker predicting the favorable efficacy of ICI treatment for patients with UC. “ in page 14 line 6-10, “These findings suggested its potential as both of diagnostic and therapeutic target in UC in addition to the predictive biomarker for efficacy of ICI treatment. However, despite extensive preclinical research, clinical translation of miRNA-based therapies has faced significant challenges and limited success. The liposomal miR-34a mimic MRX34 was the first miRNA therapeutic in a phase 1 clinical trial (NCT01829971), but development was terminated due to severe immune-mediated toxicities and patient death [51,52]. Similarly, trials of other miRNA mimics including miR-29b for fibrosis and several antimiRs targeting miR-21 and miR-122 were discontinued due to toxicity or lack of efficacy [53,54]. It has been reported that difficult challenges in advancing miRNA therapeutics include poor pharmacokinetics, off‑target effects, and inadequate delivery specificity [51-54].” in page 14 line 25-31 and page 15 line 1-4, “In our study, miR-30a-5p was significantly upregulated in urine samples from the NR group, which significantly correlated with worse PFS. In addition to the previous reports regarding the correlation between the efficacy of ICI treatment and lung cancer, miR-30a-5p could be a potential predictive biomarker for efficacy of ICI in UC patients.” in page 15 line 14-18, “Based on the result of the current study, miR-185-5p and miR-425-5p can serve as predictive biomarkers of favorable ICI efficacy in patients with UC, whereas miR-30a-5p and miR-542-3p could be associated with resistance mechanisms for ICI treatment.” in page 14 line 28-30, and it was too small cases to mention how sensitive or specificity of the identified miRNA are, so we added the sentence “This study includes just 12 patients. Further research with a larger cohort is needed to confirm the generalizability of these biomarkers. Validation in a larger cohort will be also required to confirm the predictive value of the identified urinary miRNAs and the clinical utility of our findings. However, we believe that the accumulation of such exploratory studies represents an essential first step. Building on the results of the present study, we aim to progressively increase the number of cases and ultimately conduct more extensive investigations in the future.” in page 16 line 1-7.

And we added some reference below in page 33 line 27-35 and page 34 line 1-5

  • Beg, M.S.; Brenner, A.J.; Sachdev, J.; Smith, S.; Jones, L.; Patel, R.; Lee, H.; Kim, S.; Nguyen, T.; Brown, K.; et al. Phase I study of MRX34, a liposomal miR 34a mimic, administered twice weekly in patients with advanced solid tumors (ClinicalTrials.gov identifier NCT01829971). Invest. New Drugs 2017, 35, 180–188. https://doi.org/10.1007/s10637-016-0407-y.
  • Hong, D.S.; Kang, Y.K.; Borad, M.; Smith, J.; Lee, H.; Patel, R.; Kim, S.; Nguyen, T.; Brown, K.; Davis, L.; et al. Phase I study of MRX34, a liposomal miR 34a mimic, in patients with advanced solid tumours. Br. J. Cancer 2020, 122, 1630–1637. https://doi.org/10.1038/s41416-020-0802-1.
  • van Rooij, E.; Purcell, A.L.; Levin, A.A.; Olson, E.N.; van Solingen, C.; Thum, T.; Condorelli, G.; Fortunato, O.; Hata, A.; Calin, G.A. Non coding RNA targeted therapy: a state of the art review. Int. J. Mol. Sci. 2024, 25, 3630. https://doi.org/10.3390/ijms25073630.
  • Montgomery, R.; Yu, G.; Latimer, P.; Stack, C.; Robinson, M.B.; Dalby, C.M.; Fox, S.D.; Zhang, L.; Osinska, H.; Robbins, J. MicroRNA mimicry blocks pulmonary fibrosis. EMBO Mol. Med. 2014, 6, 1347–1356. https://doi.org/10.15252/emmm.201303604.

The English could be improved to more clearly express the research.

Response; We are thankful to the Reviewer’s comment. As pointed out by Reviewer 2, we need to improve the quality of English in our manuscript. Therefore, we used the English proofreading service and modified many sentences as below.

We modified the name of Title from “Urinary microRNAs as prognostic biomarkers for predicting the effect of immune check point inhibitors in patients with urothelial carcinoma

” to “Urinary microRNAs as prognostic biomarkers for predicting the efficacy of immune check point inhibitors in patients with urothelial carcinoma”. (Page 1 line 2)

We modified the sentence from “Immune checkpoint inhibitors (ICI) have transformed urothelial carcinoma (UC) treatment, yet only a minority of patients benefit long-term due to variable response rates (<25%) and emerging resistance.” to “Immune checkpoint inhibitors (ICIs) have transformed urothelial carcinoma (UC) treatment, yet only a minority of patients benefit long-term owing to variable response rates (<25%) and emerging resistance.” in simple summary session. (Page 2 line 1-4)

We modified the sentence from “Urine samples collected prior to ICI initiation were analyzed via next-generation sequencing.” to “Urine samples collected before initiation of ICI were analyzed via next-generation sequencing.” in simple summary session. (Page 2 line 5-6)

We modified the sentence from “Patients were grouped into responders (disease control ≥6 months) or nonresponders (progression within 6 months). Findings revealed significantly elevated urinary levels of miR‑185‑5p and miR‑425‑5p in responders, correlating with enhanced overall and progression-free survival (p < 0.05). Conversely, miR‑30a‑5p and miR‑542‑3p were higher in nonresponders, indicating a potential role in immune resistance. These results suggest urinary miR‑185‑5p and miR‑425‑5p can serve as favorable predictive biomarkers for ICI response, while miR‑30a‑5p and miR‑542‑3p could signal resistance. Urine-based miRNA profiling thus represents a promising tool to inform personalized immunotherapy strategies in UC management.” to “Patients were grouped into responders (those with disease control for ≥6 months) or non-responders (those with disease progression within 6 months). Urinary levels of miR‑185‑5p and miR‑425‑5p were significantly elevated in responders, correlating with enhanced overall and progression-free survival (p < 0.05). Conversely, miR‑30a‑5p and miR‑542‑3p were higher in non-responders, suggesting a potential role in immune resistance. Therefore, These urinary miR‑185‑5p and miR‑425‑5p can serve as favorable predictive biomarkers for ICI response, while miR‑30a‑5p and miR‑542‑3p could be a signal of resistance for ICI treatment. Furthermore, urine-based miRNA profiling is a promising tool to inform personalized immunotherapy strategies in UC management.” in simple summary session. (Page 2 line 6-15)

We modified the sentence from “Background: Immune checkpoint inhibitors (ICIs) have revolutionized urothelial carcinoma (UC) treatment; however, their efficacy varies among patients. Identifying reliable biomarkers to predict response to ICIs remains a critical challenge.” to “Background: Immune checkpoint inhibitors (ICIs) have revolutionized the treatment of urothelial carcinoma (UC); however, their efficacy varies among patients. Identifying reliable biomarkers to predict response to ICIs remains challenging.” in Abstract session. (Page 3 line 2-4)

We modified the sentence from “Primary and acquired resistance to ICIs remain major challenges, necessitating the identification of predictive biomarkers and novel therapeutic targets.” to “Primary and acquired resistance to ICIs remain significant challenges, necessitating the identification of predictive biomarkers and novel therapeutic targets.”. (Page 4 line 8-10)

We modified the sentence from “The development of reliable biomarkers is essential for stratifying patients and personalizing immunotherapeutic approaches for UC.” to “The development of reliable biomarkers is essential for stratifying patients and personalizing immunotherapeutic approaches to UC.” in Introduction session. (Page 4 line 19-20)

We modified the sentence from “The JAVELIN Bladder 100 trial demonstrated an improved median OS period of 21.4 months with Ave maintenance compared with 14.3 months with best supportive care alone [10].” to “The JAVELIN Bladder 100 trial demonstrated an improved median OS of 21.4 months with Ave maintenance compared with 14.3 months with best supportive care alone [10]” in Introduction session. (Page 5 line 1-2)

We modified the sentence from “MicroRNAs (miRNAs) are small non-coding RNAs that regulate gene expression at the post-transcriptional level and play crucial roles in cancer pathogenesis, including tumor proliferation, invasion, and immune evasion [14,15].” to “MicroRNAs (miRNAs) are small non-coding RNAs that regulate gene expression at the post-transcriptional level, playing crucial roles in cancer pathogenesis, including tumor proliferation, invasion, and immune evasion [14,15].” in Introduction session. (Page 5 line 15-17)

We modified the sentence from “We identified 12 patients with advanced UC who received one of the ICIs, including Pem, Ave, and Nivo, at our institution between December 2022 and June 2023.” to “We identified 12 patients with advanced UC who received one of the ICIs (Pem, Ave, and Nivo) at our institution between December 2022 and June 2023.” in Material and Methods session. (Page 7 line 5-6)

We modified the sentence from “After explaining the study details to the patients and obtaining informed consent, urine samples were collected before the initiation of ICI treatment.” to “Urine samples were collected from patients after they received an explanation of the study and provided informed consent, prior to the initiation of ICI therapy.” in Material and Methods session. (Page 7 line 27-29)

We modified the sentence from “After discarding the supernatant, the pellet was resuspended in 250 μL of phosphate-buffered saline.” to “After the supernatant was discarded, the pellet was resuspended in 250 μL of phosphate-buffered saline.” in Material and Methods session. (Page 8 line 15-16)

We modified the sentence from “A clear difference in PFS was observed between the R and NR groups, with median PFS periods of -15.1 months in the R group and 3.7months in the NR group (Fig. S1b).” to “A clear difference in PFS was observed between the R and NR groups, with median PFS of 15.1 months in the R group and 3.7 months in the NR group (Fig. S1b) in Results session.” in Results session. (Page 11 line 6-8)

We modified the sentence from “Furthermore, after adjusting the effect of occult blood, a clear shift between R and NR was still observed in the miRNA profile along PC3 (Fig. S4b).” to “Furthermore, after the effect of occult blood was adjusted for, a clear shift between R and NR was still observed in the miRNA profile along PC3 (Fig. S4b).” in Results session. (Page 12 line 1-2)

We modified the sentence from “Progression-Free Survival Analysis We investigated the potential of the six miRNAs that were upregulated in the R group (Fig. 1b) as biomarkers. High expression levels of miR-186-5p and miR-425-5p were significantly associated with longer PFS (Fig. 2). In the high-expression group of miR-186-5p, the median PFS period was 15.0 months, compared with 3.27 months in the low-expression group.” to “PFS Analysis We investigated the potential of the six miRNAs that were upregulated in the R group (Fig. 2b) as biomarkers. High expression levels of miR-186-5p and miR-425-5p were significantly associated with longer PFS (Fig. 3). In the high-expression group of miR-186-5p, the median PFS was 15.0 months, compared with 3.27 months in the low-expression group. Similarly, the high-expression group of miR-425-5p had a median PFS of 15.0 months, whereas it was 3.27 months in the low-expression group.” in Results session. (Page 12 line 6-16)

We modified the sentence from “Similarly, the high-expression group of miR-425-5p had a median PFS period of 15.0 months, whereas it was 3.27 months in the low-expression group.” to “Similarly, the high expression group of miR-542-3p had a median PFS of 3.95 months, whereas the low-expression group had a median PFS of 15.0 months (Fig. 4).” in Results session. (Page 12 line 11-12)

We modified the sentence from “In particular, miR-186-5p is a potential tumor suppressor in certain cancers. For example, its downregulation is associated with increased cell proliferation and reduced apoptosis in breast cancer cell lines, implying that miR-186-5p may serve as a protective factor against tumorigenesis [42].” to “Particularly, miR-186-5p is a potential tumor suppressor in certain cancers. For example, its downregulation is associated with increased cell proliferation and reduced apoptosis in breast cancer cell lines, implying that miR-186-5p would serve as a protective factor against tumorigenesis [49].” in Discussion session. (Page 14 line 20-23)

We modified the sentence from “Additionally, patients with lung adenocarcinoma with low expression of miR-30a-5p exhibit increased infiltration of B cells, CD4+, T cells, and macrophases, and this infiltration of immune cells is associated with improved patient prognosis [45].” to “Additionally, patients with lung adenocarcinoma with low expression of miR-30a-5p exhibit increased infiltration of B cells, CD4+, T cells, and macrophages, and this infiltration of immune cells is associated with improved patient prognosis [56].” in Discussion session. (Page 15 line 9-12)

We modified the sentence from “Figure 2. Progression-free survival stratified by urinary miRNA levels upregulated in R. Kaplan–Meier curves of PFS in patients grouped into high and low expression by median levels of six urinary miRNAs identified as significantly upregulated in R.” to “Figure 3. Progression-free survival (PFS) stratified by urinary microRNA (miRNA) levels upregulated in responders (R). Kaplan–Meier curves of PFS in patients grouped into high and low expression by median levels of six urinary miRNAs identified as significantly upregulated in R.” in Figure Legends session. (Page 18 line 10-13)

We modified the sentence from “Figure 3. Progression-free survival (PFS) stratified by urinary microRNA (miRNA) levels upregulated in non-responders. Kaplan–Meier curves of PFS in patients grouped into high and low expression by median levels of six urinary miRNAs identified as significantly downregulated in responders.” to “Figure 4. Progression-free survival (PFS) stratified by urinary microRNA (miRNA) levels upregulated in non-responders (NR). Kaplan–Meier curves of PFS in patients grouped into high and low expression by median levels of six urinary miRNAs identified as significantly downregulated in responders (R).” in Figure Legends session. (Page 18 line 15-18)

We modified the sentence from “Gender” to “Sex” in Table 1 session. (Page 19)

Reviewer 3 Report

Comments and Suggestions for Authors

Dear Editor 

The manuscript entitled" Urinary microRNAs as prognostic biomarkers for predicting the effect of immune check point inhibitors in patients with urothelial carcinoma" evaluates the role of urinary microRNAs in prediction of efficiency of immune checkpoint inhibitors (ICIs) in urothelial carcinoma therapy. This manuscript can be considered for publication after major revision and addressing following comments point-by-point.

1-The manuscript is poorly provided in term of figures. Please prepare a schematic figure or graphical abstract which briefly illustrate overal protocol of study.

2-The conclusion is poorly written. It must be more improved 

3-Authors should add more interpretation for each section of results. 

4- Please improve resution of Figures. The legends (vertical axis) of some figures are missed. For example Fig 1c 

Comments on the Quality of English Language

The English should be polished.

Author Response

Comments to the Author

  • The manuscript is poorly provided in term of figures. Please prepare a schematic figure or graphical abstract which briefly illustrate overal protocol of study.

Response; We are thankful to the Reviewer’s comment. As pointed out by Reviewer 3, we added Figure 1 as briefly illustrating overall protocol of study in page 21 and added the sentence “We identified 12 patients with advanced UC who received one of the ICIs, including Pem, Ave, and Nivo, at our institution between December 2022 and June 2023. (Fig. 1) All patients were histopathologically diagnosed with UC. Patients who were radiologically confirmed to have progressive disease (PD) after receiving chemotherapy as the first-line therapy were treated with Pem. (Fig. 1) Patients who were radiologically confirmed to have stable disease (SD), partial response (PR), or complete response (CR) received Ave as maintenance therapy. (Fig. 1) Patients with a high risk of recurrence (pathological stage of pT3, pT4a, or pN+ and pathological stage of ypT2 to ypT4a or ypN+ for patients who received neoadjuvant cisplatin-based chemotherapy) after total cystectomy received Nivo. (Fig. 1)” in page 7 line 4-14. Please confirm the attached file (Figure 1).

  • The conclusion is poorly written. It must be more improved 

Response; We are thankful to the Reviewer’s comment. As pointed out by Reviewer 3, we need to improve the conclusion, and added the improved sentence “Conclusions

  Although ICIs have provided a breakthrough in UC treatment, their limited efficacy needs the identification of robust predictive biomarkers. We found that miR-186-5p and miR-425-5p were significantly upregulated in the urine of the R group, whereas miR-30a-5p and miR-542-3p were significantly upregulated in the urine of the NR group. These miRNAs were also significantly correlated with PFS in patients with UC who received ICI treatment. These results showed that urinary miR‑185‑5p and miR‑425‑5p can serve as favorable predictive biomarkers for ICI response, while miR‑30a‑5p and miR‑542‑3p could be a signal of resistance for ICI treatment. Urinary miRNAs can be obtained using a minimally invasive procedure and would serve as useful biomarkers for predicting the efficacy of immunotherapy in patients with UC. miRNAs could have potential for refining patient selection and guiding therapeutic decisions; however, additional large-scale studies are needed to confirm their clinical utility in the future.” in page 17 line 2-13.

  • Authors should add more interpretation for each section of results. 

Response; We are thankful to the Reviewer’s comment. As pointed out by Reviewer 3, we need to add our interpretation for each section of results, and added the improve sentence “In the current study, it was demonstrated that miR-186-5p was significantly upregulated in urine samples from the R group. Based on these previous reports and the result of our study, miR-186-5p could have potential as biomarker predicting the favorable efficacy of ICI treatment for patients with UC.” in page 14 line 6-10. and added the sentence “In our study, miR-30a-5p was significantly upregulated in urine samples from the NR group, which significantly correlated with worse PFS. In addition to the previous reports regarding the correlation between the efficacy of ICI treatment and lung cancer, miR-30a-5p could be a potential predictive biomarker for efficacy of ICI in UC patients.” in page 15 line 14-18. “Based on the result of the current study, miR-185-5p and miR-425-5p can serve as predictive biomarkers of favorable ICI efficacy in patients with UC, whereas miR-30a-5p and miR-542-3p could be associated with resistance mechanisms for ICI treatment.” in page 15 line 28-30.

4- Please improve resution of Figures. The legends (vertical axis) of some figures are missed. For example Fig 1c 

Response; We are thankful to the Reviewer’s comment. As pointed out by Reviewer 3, we need to improve Figures. We improved these Figure based on reviewer’s commnets. in page 22-23.

The English could be improved to more clearly express the research.

Response; We are thankful to the Reviewer’s comment. As pointed out by Reviewer 3, we need to improve the quality of English in our manuscript. Therefore, we used the English proofreading service and modified many sentences as below.

We modified the name of Title from “Urinary microRNAs as prognostic biomarkers for predicting the effect of immune check point inhibitors in patients with urothelial carcinoma

” to “Urinary microRNAs as prognostic biomarkers for predicting the efficacy of immune check point inhibitors in patients with urothelial carcinoma”. (Page 1 line 2)

We modified the sentence from “Immune checkpoint inhibitors (ICI) have transformed urothelial carcinoma (UC) treatment, yet only a minority of patients benefit long-term due to variable response rates (<25%) and emerging resistance.” to “Immune checkpoint inhibitors (ICIs) have transformed urothelial carcinoma (UC) treatment, yet only a minority of patients benefit long-term owing to variable response rates (<25%) and emerging resistance.” in simple summary session. (Page 2 line 1-4)

We modified the sentence from “Urine samples collected prior to ICI initiation were analyzed via next-generation sequencing.” to “Urine samples collected before initiation of ICI were analyzed via next-generation sequencing.” in simple summary session. (Page 2 line 5-6)

We modified the sentence from “Patients were grouped into responders (disease control ≥6 months) or nonresponders (progression within 6 months). Findings revealed significantly elevated urinary levels of miR‑185‑5p and miR‑425‑5p in responders, correlating with enhanced overall and progression-free survival (p < 0.05). Conversely, miR‑30a‑5p and miR‑542‑3p were higher in nonresponders, indicating a potential role in immune resistance. These results suggest urinary miR‑185‑5p and miR‑425‑5p can serve as favorable predictive biomarkers for ICI response, while miR‑30a‑5p and miR‑542‑3p could signal resistance. Urine-based miRNA profiling thus represents a promising tool to inform personalized immunotherapy strategies in UC management.” to “Patients were grouped into responders (those with disease control for ≥6 months) or non-responders (those with disease progression within 6 months). Urinary levels of miR‑185‑5p and miR‑425‑5p were significantly elevated in responders, correlating with enhanced overall and progression-free survival (p < 0.05). Conversely, miR‑30a‑5p and miR‑542‑3p were higher in non-responders, suggesting a potential role in immune resistance. Therefore, These urinary miR‑185‑5p and miR‑425‑5p can serve as favorable predictive biomarkers for ICI response, while miR‑30a‑5p and miR‑542‑3p could be a signal of resistance for ICI treatment. Furthermore, urine-based miRNA profiling is a promising tool to inform personalized immunotherapy strategies in UC management.” in simple summary session. (Page 2 line 6-15)

We modified the sentence from “Background: Immune checkpoint inhibitors (ICIs) have revolutionized urothelial carcinoma (UC) treatment; however, their efficacy varies among patients. Identifying reliable biomarkers to predict response to ICIs remains a critical challenge.” to “Background: Immune checkpoint inhibitors (ICIs) have revolutionized the treatment of urothelial carcinoma (UC); however, their efficacy varies among patients. Identifying reliable biomarkers to predict response to ICIs remains challenging.” in Abstract session. (Page 3 line 2-4)

We modified the sentence from “Primary and acquired resistance to ICIs remain major challenges, necessitating the identification of predictive biomarkers and novel therapeutic targets.” to “Primary and acquired resistance to ICIs remain significant challenges, necessitating the identification of predictive biomarkers and novel therapeutic targets.”. (Page 4 line 8-10)

We modified the sentence from “The development of reliable biomarkers is essential for stratifying patients and personalizing immunotherapeutic approaches for UC.” to “The development of reliable biomarkers is essential for stratifying patients and personalizing immunotherapeutic approaches to UC.” in Introduction session. (Page 4 line 19-20)

We modified the sentence from “The JAVELIN Bladder 100 trial demonstrated an improved median OS period of 21.4 months with Ave maintenance compared with 14.3 months with best supportive care alone [10].” to “The JAVELIN Bladder 100 trial demonstrated an improved median OS of 21.4 months with Ave maintenance compared with 14.3 months with best supportive care alone [10]” in Introduction session. (Page 5 line 1-2)

We modified the sentence from “MicroRNAs (miRNAs) are small non-coding RNAs that regulate gene expression at the post-transcriptional level and play crucial roles in cancer pathogenesis, including tumor proliferation, invasion, and immune evasion [14,15].” to “MicroRNAs (miRNAs) are small non-coding RNAs that regulate gene expression at the post-transcriptional level, playing crucial roles in cancer pathogenesis, including tumor proliferation, invasion, and immune evasion [14,15].” in Introduction session. (Page 5 line 15-17)

We modified the sentence from “We identified 12 patients with advanced UC who received one of the ICIs, including Pem, Ave, and Nivo, at our institution between December 2022 and June 2023.” to “We identified 12 patients with advanced UC who received one of the ICIs (Pem, Ave, and Nivo) at our institution between December 2022 and June 2023.” in Material and Methods session. (Page 7 line 5-6)

We modified the sentence from “After explaining the study details to the patients and obtaining informed consent, urine samples were collected before the initiation of ICI treatment.” to “Urine samples were collected from patients after they received an explanation of the study and provided informed consent, prior to the initiation of ICI therapy.” in Material and Methods session. (Page 7 line 27-29)

We modified the sentence from “After discarding the supernatant, the pellet was resuspended in 250 μL of phosphate-buffered saline.” to “After the supernatant was discarded, the pellet was resuspended in 250 μL of phosphate-buffered saline.” in Material and Methods session. (Page 8 line 15-16)

We modified the sentence from “A clear difference in PFS was observed between the R and NR groups, with median PFS periods of -15.1 months in the R group and 3.7months in the NR group (Fig. S1b).” to “A clear difference in PFS was observed between the R and NR groups, with median PFS of 15.1 months in the R group and 3.7 months in the NR group (Fig. S1b) in Results session.” in Results session. (Page 11 line 6-8)

We modified the sentence from “Furthermore, after adjusting the effect of occult blood, a clear shift between R and NR was still observed in the miRNA profile along PC3 (Fig. S4b).” to “Furthermore, after the effect of occult blood was adjusted for, a clear shift between R and NR was still observed in the miRNA profile along PC3 (Fig. S4b).” in Results session. (Page 12 line 1-2)

We modified the sentence from “Progression-Free Survival Analysis We investigated the potential of the six miRNAs that were upregulated in the R group (Fig. 1b) as biomarkers. High expression levels of miR-186-5p and miR-425-5p were significantly associated with longer PFS (Fig. 2). In the high-expression group of miR-186-5p, the median PFS period was 15.0 months, compared with 3.27 months in the low-expression group.” to “PFS Analysis We investigated the potential of the six miRNAs that were upregulated in the R group (Fig. 2b) as biomarkers. High expression levels of miR-186-5p and miR-425-5p were significantly associated with longer PFS (Fig. 3). In the high-expression group of miR-186-5p, the median PFS was 15.0 months, compared with 3.27 months in the low-expression group. Similarly, the high-expression group of miR-425-5p had a median PFS of 15.0 months, whereas it was 3.27 months in the low-expression group.” in Results session. (Page 12 line 6-16)

We modified the sentence from “Similarly, the high-expression group of miR-425-5p had a median PFS period of 15.0 months, whereas it was 3.27 months in the low-expression group.” to “Similarly, the high expression group of miR-542-3p had a median PFS of 3.95 months, whereas the low-expression group had a median PFS of 15.0 months (Fig. 4).” in Results session. (Page 12 line 11-12)

We modified the sentence from “In particular, miR-186-5p is a potential tumor suppressor in certain cancers. For example, its downregulation is associated with increased cell proliferation and reduced apoptosis in breast cancer cell lines, implying that miR-186-5p may serve as a protective factor against tumorigenesis [42].” to “Particularly, miR-186-5p is a potential tumor suppressor in certain cancers. For example, its downregulation is associated with increased cell proliferation and reduced apoptosis in breast cancer cell lines, implying that miR-186-5p would serve as a protective factor against tumorigenesis [49].” in Discussion session. (Page 14 line 20-23)

We modified the sentence from “Additionally, patients with lung adenocarcinoma with low expression of miR-30a-5p exhibit increased infiltration of B cells, CD4+, T cells, and macrophases, and this infiltration of immune cells is associated with improved patient prognosis [45].” to “Additionally, patients with lung adenocarcinoma with low expression of miR-30a-5p exhibit increased infiltration of B cells, CD4+, T cells, and macrophages, and this infiltration of immune cells is associated with improved patient prognosis [56].” in Discussion session. (Page 15 line 9-12)

We modified the sentence from “Figure 2. Progression-free survival stratified by urinary miRNA levels upregulated in R. Kaplan–Meier curves of PFS in patients grouped into high and low expression by median levels of six urinary miRNAs identified as significantly upregulated in R.” to “Figure 3. Progression-free survival (PFS) stratified by urinary microRNA (miRNA) levels upregulated in responders (R). Kaplan–Meier curves of PFS in patients grouped into high and low expression by median levels of six urinary miRNAs identified as significantly upregulated in R.” in Figure Legends session. (Page 18 line 10-13)

We modified the sentence from “Figure 3. Progression-free survival (PFS) stratified by urinary microRNA (miRNA) levels upregulated in non-responders. Kaplan–Meier curves of PFS in patients grouped into high and low expression by median levels of six urinary miRNAs identified as significantly downregulated in responders.” to “Figure 4. Progression-free survival (PFS) stratified by urinary microRNA (miRNA) levels upregulated in non-responders (NR). Kaplan–Meier curves of PFS in patients grouped into high and low expression by median levels of six urinary miRNAs identified as significantly downregulated in responders (R).” in Figure Legends session. (Page 18 line 15-18)

We modified the sentence from “Gender” to “Sex” in Table 1 session. (Page 19)

Round 2

Reviewer 1 Report

Comments and Suggestions for Authors

Authors ameliorated the manuscript with reviewer’s comments.

Reviewer 3 Report

Comments and Suggestions for Authors

The manuscript is improved as well and can be published in present form.